# Kinbank: A global database of kinship terminology

Sam Passmore[1,2,3]*, Wolfgang Barth[4], Simon J. Greenhill[5,6], Kyla Quinn[4], Catherine Sheard[2,7], Paraskevi Argyriou[8], Joshua Birchall[9,10], Claire Bowern[11], Jasmine Calladine[2,12], Angarika Deb[2,13], Anouk Diederen[14], Niklas P. Metsäranta[15], Luis Henrique Araujo[16], Rhiannon Schembri[17], Jo Hickey-Hall[2], Terhi Honkola[2,15], Alice Mitchell[2,18], Lucy Poole[2], Péter M. Rácz[2,19], Sean G. Roberts[2,20], Robert M. Ross[21], Ewan Thomas-Colquhoun[2], Nicholas Evans[1,4]*, Fiona M. Jordan[2]*

1 Evolution of Cultural Diversity Initiative (ECDI), Australian National University, Canberra, ACT, Australia, 2 Department of Anthropology and Archaeology, University of Bristol, Bristol, United Kingdom, 3 Faculty of Environment and Information Studies, Keio University, Fujisawa, Japan, 4 ARC Centre of Excellence for the Dynamics of Language (CoEDL), Australian National University, Canberra, ACT, Australia, 5 School of Biological Sciences, University of Auckland, Auckland, New Zealand, 6 Department of Linguistic and Cultural Evolution, Max Planck Institute for Evolutionary Anthropology, Leipzig, Germany, 7 School of Earth Sciences, University of Bristol, Bristol, United Kingdom, 8 School of Biological and Behavioural Sciences, Queen Mary University of London, London, United Kingdom, 9 Museu Paraense Emílio Goeldi, Belém, Pará, Brazil, 10 Department of Linguistics, The University of New Mexico, New Mexico, United States of America, 11 Department of Linguistics, Yale University, New Haven, Connecticut, United States of America, 12 Department of Human Behavior, Ecology and Culture, Max Planck Institute for Evolutionary Anthropology, Leipzig, Germany, 13 Department of Cognitive Science, Central European University, Vienna, Austria, 14 Max Planck Institute for Psycholinguistics, Nijmegen, Netherlands, 15 Department of Finnish, Finno-Ugrian, and Scandinavian Studies, University of Helsinki, Helsinki, Finland, 16 Department of Linguistics, Universidade Federal do Pará, Belém, Pará, Brazil, 17 Research School of Biology, Ecology, and Evolution, Australian National University, Canberra, ACT, Australia, 18 Institute for African Studies, University of Cologne, Cologne, Germany, 19 Cognitive Science Department, Budapest University of Technology and Economics, Budapest, Hungary, 20 School of English, Communications and Philosophy, Cardiff University, Cardiff, United Kingdom, 21 School of Psychological Sciences, Macquarie University, Sydney, NSW, Australia

* Samuel.passmore@anu.edu.au (SP); Nicholas.evans@anu.edu.au (NE); Fiona.jordan@bristol.ac.uk (FMJ)

**Data Availability Statement:** The Kinbank database is freely accessible at https://zenodo.org/record/6471794, as are details of how each individual dataset can be cited (S2 File). The data is interactively available at www.kinbank.net.

## Abstract

For a single species, human kinship organization is both remarkably diverse and strikingly organized. Kinship terminology is the structured vocabulary used to classify, refer to, and address relatives and family. Diversity in kinship terminology has been analyzed by anthropologists for over 150 years, although recurrent patterning across cultures remains incompletely explained. Despite the wealth of kinship data in the anthropological record, comparative studies of kinship terminology are hindered by data accessibility. Here we present Kinbank, a new database of 210,903 kinterms from a global sample of 1,229 spoken languages. Using open-access and transparent data provenance, Kinbank offers an extensible resource for kinship terminology, enabling researchers to explore the rich diversity of human family organization and to test longstanding hypotheses about the origins and drivers of recurrent patterns. We illustrate our contribution with two examples. We demonstrate strong gender bias in the phonological structure of parent terms across 1,022 languages, and we show that there is no evidence for a coevolutionary relationship between cross-cousin marriage and bifurcate-merging terminology in Bantu languages. Analysing kinship data is

**Funding:** FMJ received funding from the European Research Council (Starting Grant VARIKIN ERC-Stg-639291). NE received funding from the Australian Research Council Center of Excellence for the Dynamics of Language (Grant CE140100041). JB and FMJ received funding from the British Academy (International Partnership Mobility Grant 160281). The funders had no role in study design, data collection and analysis, decision to publish, or preparation of the manuscript.

**Competing interests:** The authors have declared that no competing interests exist.

notoriously challenging; Kinbank aims to eliminate data accessibility issues from that challenge and provide a platform to build an interdisciplinary understanding of kinship.

## Introduction

Human kinship organization is remarkably diverse. Our patterns of caring for each other, finding partners, and cooperating with relatives show more variation than any other species on the planet, despite kinship being anchored in biological and social constraints [1]. Kinship relations in human communities are created through both basic reproductive processes and the social making of family ties which, importantly, are transmitted through language and culture [2]. Central to social organization, kinship has influenced many aspects of our evolutionary history including the distribution of linguistic and genetic diversity [3], technology [4], the likelihood of external warfare [5, 6], and is thought to be the factor behind purported psychological "WEIRD-ness" observed in western societies [7]. Anthropological studies of kinship have traditionally encompassed marriage customs, the tracing of descent and community relations, the jurisdiction of rights and responsibilities in offspring, and the variety of residential groupings for family, with existing calls to use quantitative and evolutionary methods to understand kinship diversity, and its role in cultural evolution and linguistic change [8, 9]. In the last half-century this remit has broadened, including e.g. new reproductive technologies, single-parent families, LGBT+ kinship, and different kinds of relatedness [10, 11]. While research foci change, one consistent strand has been the linguistic denotation and organization of family members in *kinship terminology*: the patterned vocabulary of words for kin.

This paper introduces Kinbank, a free and open-source database that centralizes and systematizes global cross-cultural data on kinship terms of 1,229 spoken and signed languages, providing an accessible and comprehensive resource for documenting and analyzing kinship diversity. The organization of kinship terminology is remarkably variable across languages. Different languages distinguish relationships by using distinct terms, or they conflate the relationship by using the same term. For example, in Lau Fijian (Fiji) a woman refers to her male sibling using the same word that he refers to her (*weka* i.e. 'opposite-sex sibling'); Ngiyambaa (Australia) only has one term for children, unlike son and daughter in English; and Dutch (Netherlands) groups parent's siblings children (cousins), and sibling's children (nieces and nephews) together under *neef* (m) and *nicht* (f).

This structured variation in kinship terms has preoccupied anthropologists for 150 years [12]. Virtually all ethnographic and descriptive linguistic scholars in the 20[th] century elicited kinterms from speakers in the communities they studied. As a result, kinship terminology is richly documented. However, these data are scattered. Terms have been collected in comparative surveys [9], documented within ethnographies [13], included as word lists within grammars [14], or scattered through dictionaries and specialized articles. Bringing these data together is important, because kinship terminology studies have made valuable contributions to our understanding of social structure. For example, studies have shown how linguistic denotation indicates group marriage norms [15] and how those rules are manipulated for economic [16], or reproductive benefit [17, 18]. Research has used kinterms to identify cross-cultural differences in parental care roles [19] and revealed logical structures in terminology [20, 21]. Furthermore, variations in kinship terminology have been used to infer the structures of early human society [22] and societies in prehistory [23–25].

In specialist study, kinship terminology has been a fruitful avenue for understanding contemporary and historical relationships between language and behavior. For example, the adoption of novel algebraic methods to transcribe and interpret kinship terminologies have provided valuable insights to the intersection of genealogical and cultural understandings of kinship [26]. Similarly, applying optimality theory has reframed kinship terminology diversity within a domain of rules and constraints that can generate cultural variability [1]. Despite the value of these approaches, there is an increasing disconnect between the field's theoretical advancement and the data routinely used by other social sciences. The specialist knowledge needed to engage with cutting-edge kinship scholarship has meant work in fields such as psychology, linguistics, and wider anthropology largely rely on 70-year-old typologies [27], which lack cross-cultural validity [28]. This disconnect perpetuates ideas that existing knowledge does not support, such as the relationship between kinship typology and social norms [29]. Kinbank has the potential to re-engage diverse disciplines with the study of kinship terminology by offering streamlined access to data and sources, allowing specialists to quickly expand and generalize their findings, and by centralizing key new data in a single resource.

Since the turn of the 21st century, fields such as anthropology, psychology, and linguistics have taken a (re)turn to the empirical analysis of kinship systems [1, 8, 27, 30]. Cognitive models of kinship terminology highlight strategies to uncover universal processing principles [1, 31, 32], and social categories of kinship are used to prescribe cultural patterns of behavior [33, 34]. Such studies rely on kinship terminology, but have been restricted to existing aggregated data [31], or sourcing their own terminology [32], and would be improved or aided by larger datasets. Kinship is also becoming increasingly central to economic historians, who have used kinship terminology structure to infer broader patterns of social structure [7, 35]. With access to broader data, members of the Kinbank project have: demonstrated links between linguistic and social organization [36], and the regional variation in these relationships [29]; shown the modularity of Pama-Nyungan kinship terminology and quantified the connections between modules [37]; presented a collection of terms for languages whose kinship systems had never been fully described alongside new methods for Tupian and Cariban languages [38]; used corpus-based studies to show the difference in cultural evolutionary changes between kinship words and basic vocabulary [39]; used field studies to revive research on the acquisition of kinship language and concepts by children [40, 41]; and performed multiple studies showing how kinship finds its way into core grammar [42, 43].

Kinbank centralises terminology from approximately 15% of known language diversity, at least 3.28 billion speakers (47% of the speakers in Ethnologue), including speakers for 96% of the territories covered in Ethnologue, and at least 20% of every continent [44]. Kinbank contains the largest and broadest collection of terminology available, offering a platform to investigate variation in kinship terminology. Researchers can leverage Kinbank to further explore the topics above or many other questions regarding kinship and kinship terminology. We offer two examples of analyses. First, we show a global analysis on forms of kinterms, testing whether we observe gender bias in the phonology of parental terms. Second, we use phylogenetic techniques to test for a co-evolutionary relationship between kinship terminology structure and patterns of cross-cousin marriage. These examples exemplify the types of methodological and theoretical approaches researchers can address using the Kinbank dataset. For example: is there any generalisable relationship between the behavioural structures and social norms of kinship and kinship terminology? What are the unattested categories of kin and what does that tell us about the constraints of language? How learnable are different terminological systems? The questions we can ask of kinship terminology are plentiful. With a comparative database, they are now also testable.

## The Kinbank database

### Data, sampling, and database structure

Kinbank provides a digitized, open-access, and global database of kinship terminologies, resulting from the international, multi-funder collaboration of four aligned research projects. They are: Parabank based at the ARC Centre of Excellence for the Dynamics of Language, Australian National University; Varikin at the University of Bristol; MPEGKin at the Museu Paraense Emílio Goeldi; and Kinura at the University of Helsinki. The Kinbank database is freely accessible at https://zenodo.org/record/7232746, as are details of how each individual entry or dataset can be cited (S1 Table in S1 File). Kinbank is subject to a Creative Commons Attribution Non Commercial 4.0 International licence.

### The etic approach

Capturing the global variation in kinship terminology requires a framework from which to compare languages. While much of the recent academic literature about kinship terminologies is written in English, the variety of categorizations found cross-linguistically means that using English kinship terms to describe cross-cultural variation would be scientifically inaccurate and ethnocentric. Historically, there have been two approaches to kinship terminology: emic (language-internal logic) and etic (objective language-independent grid for comparison) [45]. The emic approach seeks to unlock the inner logic of a language's kinship terms by taking locally meaningful categories of terms as a fundamental unit of interest. The etic approach relies on a language independent yardstick, often in the form of a genealogical grid of *kin types* (a genealogical position in the etic grid), which *kinterms* (the word used to describe one or more genealogical positions) are laid on top of. For example: the Kayardild (Australia) kinship terms *kularrint* and *duujint* would be emically described as "opposite-sex sibling" and "younger same-sex sibling", but *kularrint* would be etically described as equating to man's female sibling and woman's male sibling, and *duujint* would be man's younger male sibling, and woman's younger female sibling. The emic approach provides a succinct description of the relationship and how the terms fit together within the language, but is difficult to apply cross-linguistically (e.g. opposite-sex sibling does not apply in an English terminology). The formulations used in an etic approach are less elegant, but we can more easily apply the category of man's younger male sibling to other languages (*brother* in English).

The goal of this database is to facilitate high-level cross-cultural comparison by providing the first open standardized base from which to compare kinship terms across languages, so Kinbank assumes an etic and genealogical approach. The etic approach provides a fine-grained and language-independent yardstick for the set of kin types over which each kin term can range, enabling us to quantify the similarity and differences across languages [46]. Here are three examples of how this could apply to kinship terminology: 1) By measuring how far particular terms extend over etically defined referents like father, father's brother, and mother's brother, we can readily measure the extent of referential categories and compare their semantics across societies: e.g. Does *uncle* for parent's male siblings in English cover the same relatives as *oba* in Japanese (yes) or *kakuju* in Kayardilt (no)?. 2) We may ask questions about the similarity of referential ranges in different subsets of kinterms: e.g. Do parental terms tend to be extended to parents' siblings more often than sibling terms to parents' siblings' children? And 3) we can also explore the structural similarity of kinship terminologies across languages [28]. Using an etic grid presents the opportunity to ask more of these types of questions and to draw macro- and cross-cultural conclusions. This has been done in previous approaches, such as when Nerlove and Romney [47] used an etic grid to determine 12 types of sibling

terminologies, representing 98% of the observed languages. These authors established a design space of 4,140 possible sibling terminologies, derived from an etic set of eight sibling categories derived from three rules of distinction (gender of sibling, gender of speaker, and relative age). Using Kinbank's etic structure and advances in statistical methods, we can extend this approach to larger subsets of kin and more complex methods of comparison (e.g., [28]).

## The Kinbank sample

At time of publication, Kinbank holds 210,903 different data points across 1,229 languages (Fig 1). The collaboration between Parabank, Varikin, MPEGKIN, and Kinura has resulted in a database with a broad global sample, coupled with focused sampling from specific language families. Each language is linked to a stable and unique linguistic identifier "Glottocode" that is commonly used to link language and cultural data to other databases [48]. By indexing on source, we can also separate terminologies within a language across sources, which allows analyses to measure concordance across sources, track change over time, or identify within-language variation. Each project had its own sampling objective, which we describe below.

Parabank collected kinship terminologies opportunistically, including the digitisation of the terminologies from Morgan's 1871 landmark global kinship survey, *Systems of Consanguinity and Affinity* [12]. The global nature of the Parabank sample means the database holds terminologies from societies with a range of ecological pressures and kinship structures, allowing us to explore convergent patterns of terminology (e.g. [28] across the global sample.

Varikin focused on sampling languages that are paired both with a dated language phylogeny (e.g. [49] and with existing anthropological databases (e.g. D-PLACE [50]). The most sampled language families by Varikin wereAustronesian (n = 375), Atlantic-Congo (117), Indo-European (105), and Pama-Nyungan (104). This sampling strategy allows for a range of phylogenetic analyses that control for patterns of autocorrelation that might occur through descent [51, 52]. The focus on societies within the Ethnographic Atlas additionally allows kinship terminology data to be connected to demographic and health surveys [53].

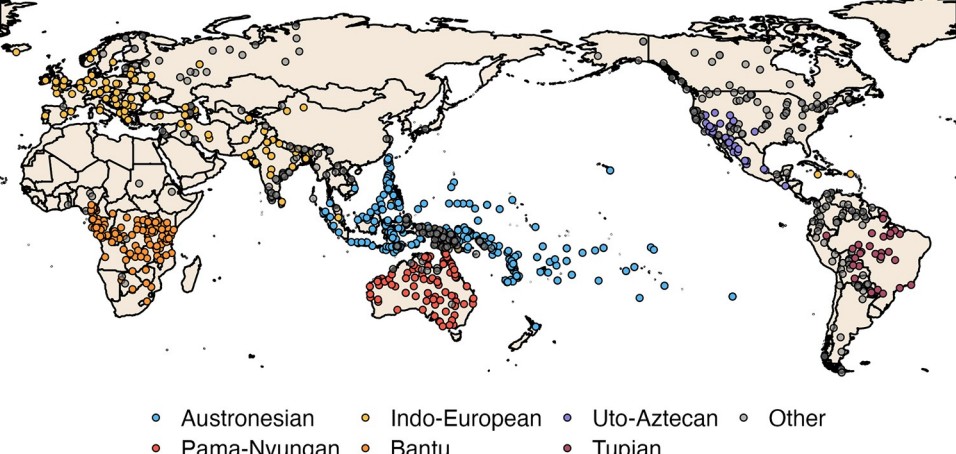

**Fig 1. Locations of languages in Kinbank.** Each point indicates a unique language variety and is centered on the geographical center-point of the area where the speakers live when the data were recorded, but may also indicate a historical location, the demographic center-point or some other representative point. Colored points indicate languages from the 7 language families mentioned in text: Austronesian (light blue; n = 377); Pama-Nyungan (red; n = 105); Indo-European (yellow; n = 106), Bantu (orange; n = 113), Uralic (purple; n = 25), Tupian (maroon; n = 29); Cariban (pale blue; n = 7) and other languages (grey).

MPEGKin compiled data on two large language families of South America, Tupian and Cariban, sampling as densely as possible across the major branches of the families given the available ethnographic and linguistic documentation. Beyond making new data available for these lesser-known languages [38], MPEGKin was designed to support quantitative comparative studies and traditional linguistic reconstruction with these language families (e.g., [54, 55]).

Kinura collected kinship terminologies of all the main groups of the Uralic family. The sampling focused on matching the kin term data with the speaker groups included in the genetic study by Tambets et al. in 2018 [56].

Focused sampling for languages attached to existing cultural and linguistic databases (e.g. D-PLACE [50]) and to computational language phylogenies (e.g. [57]), allows evolutionary analytic approaches to this important cultural domain. The combination of focused regional samples and broad global coverage will allow future research to compare the patterns and correlates of diversity across different scopes. Kinbank is citable as a whole and as separate regional projects (see S1 Table in S1 File).

## Concepts in Kinbank

The primary search criteria for Kinbank were a core set of 115 kin types (88 genealogical kin and 27 kin by marriage (affine); available in S2 Table in S1 File). The core set of kin types encompass parents, siblings, and children; up to grandparents and down to grandchildren, and then from parents to their siblings and parents' siblings' children. We limited the set to two generations above and below the ego (grandparents to grandchildren). This is a commonly discussed range in the kinship literature, but it also represents the most common set of generations that exist during ego's lifetime, which is four generations with a generation of between 20–30 years. Not all generations necessarily exist at the same time, e.g., ego might have grandparents when they are young, and grandchildren when they are old. It is uncommon for five generations to co-exist. Many languages have separate terms depending on the gender of the speaker, and we collected all such terms where available. We aimed to collect affinal terms for spouses, spouses of siblings, and the spouses' nuclear family. Within ego's generation (ego being the central figure from which relationships stem) and ego's parents' generation, kin types are also distinguished by relative age, and age of linking relative, where appropriate. This set is derived from a genealogical grid of relatives and aims to capture a globally recurrent set of cross-culturally valid kin-members.

The dataset contains terms of reference (the answer to: "who is this person to you"), as opposed to terms of address (the answer to: "what do you call this person") (48). Reference terms are more commonly the focus of anthropological data collection, since these designate formal categories of relationship (e.g. "father" vs "dad"), and are less prone to politeness effects which often extend the range of kinterms (e.g. by employing "uncle" / "aunt" to address non-related elder men and women as a term of respect). Terms of address were recorded when they were easily available but were not the focus of this collection.

The set of individuals designated by kinterms is culturally variable [58]. Some communities have a restricted set of kinterms, and others an expanded set extending beyond the established genealogical grid [59]; however, since the unit of the database is the kinterm, the database is flexible to account for different emic characterizations of kin relations. In the case of a restricted set, cells that are not required are left empty. This presents the difficult challenge of knowing whether a term is absent because this relative is not considered a kin member or because it was not recorded in the source. The difference between a language lacking a term for a particular kin type (true absence), and available records of a language lacking particular

terms (data uncertainty), is important. Comparisons with closely related languages, the use of multiple sources, and iterative, community-elicited data collection can help to narrow this uncertainty in future.

## Collection procedure

Data are entered into Kinbank as either primary or secondary data, and is affiliated with a source reference detailing where the data came from. The data are most commonly secondary, and transcribed by coders into the Kinbank collection template. Secondary sources ranged from ethnographies and grammars, to simpler descriptions like wordlists. In the majority of cases, secondary sources describe kinterms etically following the tradition laid out by Morgan (1871). The etic descriptions usually align with the list of 115 core kin types. In these instances, kin term collection is a matter of copying terms to the appropriate place. It is common for source categories to be less refined than Kinbank categories. In this case, coders would apply the term to all relevant categories. For example: the category father's brother is commonly used in secondary sources, but this aggregates four Kinbank categories (father's elder brother, father's younger brother, and whether the speaker is male or female). A coder would apply the term to all four of these categories in the absence of further information. In the rare instance that kinterms were not etically described (often using English nomenclature), coders would search the text for context and confirmation that these categories align, and record any inferences in the notes column for each kin term. Coders were also encouraged to flag where original sources may have errors.

Kin terms are transcribed as closely as possible to their source form. Since terminologies are predominantly collected by anthropologists and not linguists, sources are primarily in Roman script (as opposed to other standardized orthographies) and can contain transcription inconsistencies across languages. Kinbank makes no judgments on which transcriptions are correct or incorrect, besides obvious typos. The database accommodates phonetic transcriptions in the International Phonetic Alphabet where these are available. Some sources recorded terminology using Cyrillic; in this case kinterms are transliterated into Roman characters, with the original form in an adjacent column. This approach means that the copying is consistent within source but may not be consistent across sources or languages.

Some data was elicited from native speakers or academics actively working with a particular language. In this case the person(s) who offered the terms is listed as the source. When data was collected from native speakers, the fieldworker would use various methods to collect kinterms. For example: the fieldworker would have informants explain their genealogy, eliciting kinterms through this process, coupled with participant observation to observe the kinterms being used in context. Academics actively studying a language were provided with the template sheet and a description of the task and asked to fill in the categories with any necessary comments or adjustments, using the methods above. When there was specific context, meaning, or otherwise interesting information for a particular kinterm it was recorded in the notes. Some sources (speakers or secondary) may offer many more kinterms that apply beyond the 115 core types. In this instance, we expand the parameter list to accommodate the additional kinterms (see S2 Table in S1 File). In total there are 940 kin types within Kinbank, but many of these are sparsely populated. Some languages also contain multiple terms for a particular kin type. In this instance each kin term is entered separately.

## Kinbank database format

Data are stored and distributed in the Cross-Linguistic Data Format (CLDF; [60]), which is a flat structured database stored in comma-separated value (CSV) files (Table 1), and built with

**Table 1. Descriptions of the tables used in Kinbank and the data they contain.** An extended description with additional structuring files is given in S3 Table in S1 File.

| File | Description |
|------|-------------|
| forms.csv | This contains the forms, or kinterms, for each language. This file links to the languages.csv file by the Languages_ID column, parameters.csv file by the Parameters_ID column, and the sources.bib file by the sources_bibtex column. |
| languages.csv | This file contains metadata on each language (Name, Glottocode, ISO639P3 code, Macroarea, Latitude, Longitude, and Language Family) |
| parameters.csv | This file contains descriptions of each kin type used in the dataset. |
| sources.bib | A bibtex file containing information on the sources used in Kinbank. |

the affiliated tools CLDFbench [61]. This format is easily importable into common analysis tools like Microsoft Excel, and data analysis languages like R or Python.

It is common for languages to have multiple terms recorded for some kin types. Due to the combination of form and kin type, the database can accommodate multiple terms per kin type. Multiple terms for a single kin type may occur for a number of reasons: borrowed and native terms coexisting, dialectal differences, different registers, or simply language flexibility [62]. The database offers no judgement as to the specific cause. Again, users can refer to the original source(s) for further information.

Kinbank was designed to be corrected and expanded over time. The 'living' datasets are stored at github.com/kinbank. By using Github to store the living datasets, users can raise issues with existing data, propose corrections, and upload new languages. All changes are reviewed by Kinbank authors. New languages can be added by filling in the template document of core kin types found on the homepage of each repository. Living database are not static. To ensure stability to researchers, the datasets are periodically versioned on Zenodo.

## Inter-rater reliability

Forty-four languages were collected independently by both ANU and Bristol teams and used to determine the level of intercoder reliability. A major avenue for error is when kinship terms are collected for one language but two different sources disagree on kinterms. We focus on ensuring the structural paradigm of a particular kinship terminology is consistent (i.e. that all parent's female siblings are syncretized to "aunt" in English, not whether one source specifies "aunt" or "aunty"); inter-rater reliability is based on this benchmark. To compare collections, we used a structural similarity kappa value, which compares the pattern of syncretism within each collection, accounting for similarity by chance. Across all compared relationships, we obtained a structural similarity kappa value of 0.80. Calculation and discussion of interrater reliability is available in supplementary material and S4 Table in S1 File.

## Analysis of Kinbank data

The Kinbank collection of 1,229 languages offers a large digitised sample of kinship terminology in an accessible format. The collection has 887 languages that contain all grandparent terms, 864 languages with all sibling terms, 728 languages with all terms for parents and parent's siblings, 604 languages with all nibling terms, and 506 languages with all sibling and cousin terms. The most frequently collected kinterms are all nuclear family terms: Father (n = 1,197), Mother (1,192), younger brother (1,161), elder brother (1,149), elder sister (1,138), and younger sister (1,134). Closely followed by father's father (1,110) and mother's father (1,119). The least common terms, from within the core set, are a combination of affinal and

cousin terms: wife's sister (545), father's brother's daughter (552), husband's sister (552), husband's brother (554), and mother's sister's daughter (556). On average, languages have 23 terms: 2 grandparent terms, 3 sibling terms, 5 terms for parents and parent's siblings, 4 nibling terms, and 6 sibling and cousin terms. Averages only tell us part of the picture however, as languages can have as low as one kin term for each set (with the exception of parents and parent's siblings which has a minimum of two) or multiple terms for every category. Although languages range significantly in the number of kinterms they use, some syncretisms are "socially unthinkable" [63, 64]. The law of collaterality states that we cannot observe syncretisms between kin that are linked by a member of the opposite sex, without other parallel sycrentisms occurring, a rule which is abided by in all Kinbank languages. For example: Kinbank contains no languages which have a term for father and for mother's brother, but not father's brother. Similarly, Kinbank contains no languages that have distinct terms for father's mother, or mother's father, with a single term for all other grandparents.

Below we provide two examples of how researchers can use Kinbank's features and data to provide powerful insights to cultural change. First, we investigate the phonological structure of global terms for "mother" and "father" to show strong gender bias with regard to phonological form. Second, we integrate Kinbank with anthropological data and find no evidence to support a coevolutionary relationship between kinterms and marriage systems in Bantu languages.

**Example 1: Are you my mama? Gender bias in the phonology of parental terms.** The global recurrence of certain sounds in parental terms (e.g. [ma], [pa]) among geographically distant and historically unrelated languages is hypothesised to stem from the constraints on early baby babbling [65]. A babbling theory for parental term similarity was first identified by Murdock [66] as a statistical regularity, and then subsequently theorized by [65] to recur because of the maximal phonetic contrasts the sounds make. The combination of a stop or nasal, followed by a low vowel, creates the largest contrast amongst the sound's babies are capable of making and are recognizable, distinguishable, and identifiable noises. The bias towards these perceivable signals of early communication then further evolves to become codified as words for parents in spoken language, as opposed to its babbled form.

An extension of this theory suggests that mother terms are more likely to start with [ma] because of the sound relationship to breastfeeding [65, 66]. The bilabial nasal sound [m] is putatively an anticipatory murmur; the [a] sound is created by the baby's mouth opening preparing to breastfeed. The onomatopoetic link to breast-feeding is hypothesized to produce a phonological gender bias. Although early descriptive statistics support this conclusion [66], recent research on Australian languages has shown that parental terms with initial [ma] sounds frequently refer to father, not mother; furthermore *mama* is a reconstructable proto-form for "father" in Pama-Nyungan languages [67, 68]. [67] also show that other nasal consonants such as [ŋ] (velar nasal) are commonly used for mother, which are also compatible with the breastfeeding hypothesis. The regional and historical relationship between [ma] and "father" in Australia highlights the importance of accounting for the interdependence between languages when looking at cross-linguistic patterns. Had the sample been biassed towards Australian languages without a historical control, we would conclude that most languages use [ma] for father, but a phylogenetic control accounts for this bias through shared evolutionary history [52]. Here, we ask whether there is a gender bias in the phonology of parental terms, using phylogenetic approaches to explicitly model and control for the historical relationships between languages. We code the first syllable of 3,068 parental kinterms from 1,022 languages for a consonant and vowel, following categories established in [69], and model their relationship to the referent using a phylogenetically-controlled repeated measures multilevel Bayesian logistic regression (See methods for more detail).

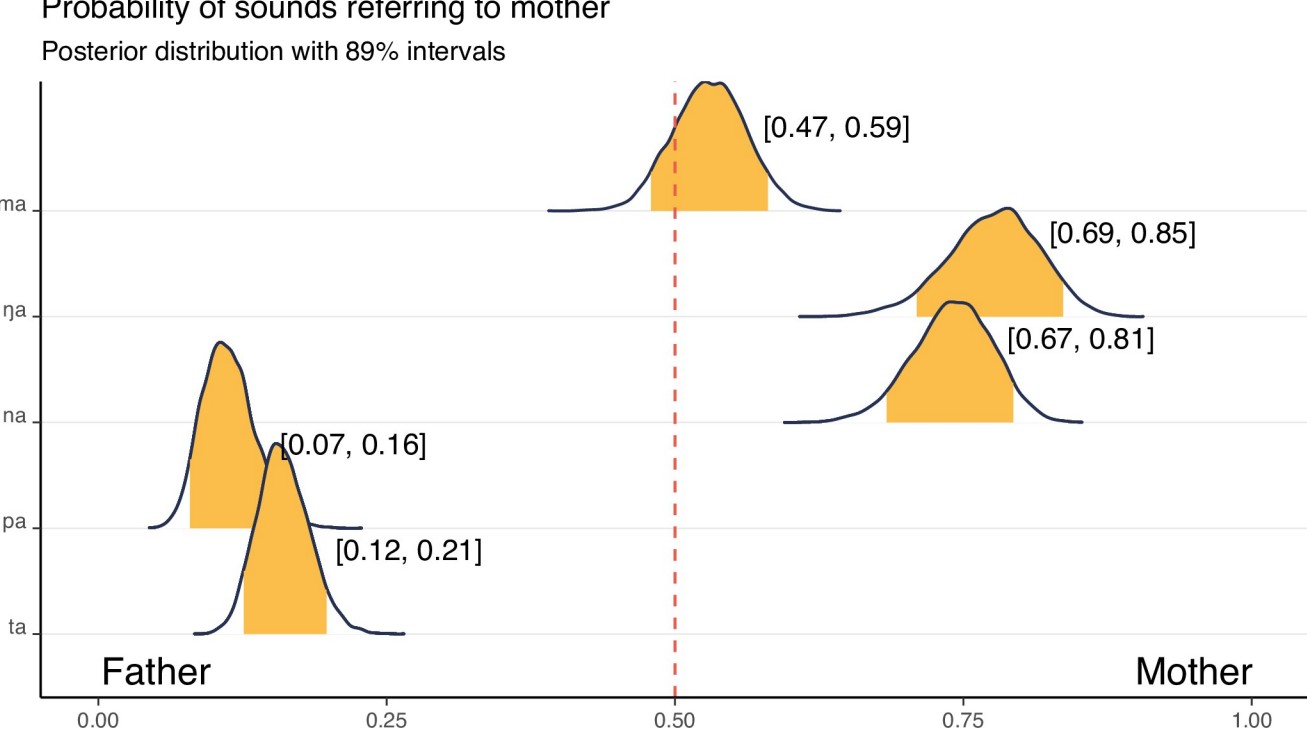

**Fig 2. Probability of consonant and vowel combinations used in words referring to mother.** Each row shows the probability density estimate from the model posterior. The colored sections and annotated numbers show 89% highest probability density intervals. Intervals that contain 0.5 have no statistical effect (since there is a 50:50 chance the sound refers to mother or father). Results show words with a root syllable of [ŋa] and [na] are more likely to refer to mother and [pa] and [ta] to refer to father. [ma] words are predicted to be used equally between mother and father words.

**Example 1: Results.** The model shows that consonant use is a strong predictor of parental referent. Vowels offer little predictive value. Since consonant-vowel combinations are the real-world depiction of these sounds, we calculate the probability of theoretically implicated, and contrastive, consonant-vowel pairs from the model posterior: [ma], [na], and [ŋa] (common mother sounds), and [pa], and [ta] (common father sounds; since vowels have little influence we only use [C-a] in the examples; Fig 2). The model only analyses "mother" or "father" words: mother words are coded as 1. Since the only other possibility is for a word to refer to father, probabilities close to zero indicate the likelihood a sound refers to father. There is positive evidence for [ŋa] and [na] more commonly referring to mother, and that [pa] and [ta] more commonly refer to father. Notably, there is no evidence that [ma] sounds refer preferentially to mother or to father. A summary of the model and all effects are available in S5 Table in S1 File.

**Example 1: Discussion.** Our analysis has shown that the bilabial-low vowel combination [ma], which is often thought to be linked to mothers, is actually equally linked to fathers. On the other hand, we present strong evidence in a global sample for [ŋa] and [na] sounds aligning exclusively with mother terms, as reported by [67, 68] for Australian languages. Closer inspection of Murdock's tabulations shows very few languages using velar nasal sounds in parental terms, in stark contrast to their relatively high frequency in our broader sample. This difference highlights the importance of wide language documentation, and an understanding of the full range of linguistic diversity before drawing premature "universal" conclusions [70]. Kinbank provides us with a broader sample of diversity, which reveals a new understanding of gender bias in parental term phonology. With a broader sample comes the possibility that languages are similar because of historical relatedness, rather than an external factor. Here, we

have used global linguistic relationships to control for the possibility of auto-correlation within our sample. A puzzle yet to be solved is whether there is a causal reason for the relationship between [ta] and [pa] and father terms, or whether the bias is a result of phonological dispersion i.e., maximal differentiation as a phonological driver for parental terms.

**Example 2: Does crossness indicate marriage preferences?.** Within anthropology, the structure of kinship terminology is thought to reflect patterns of social structure [9]. A common example is the linguistic denotation of marriage taboos, indicating which kin are marriageable and which are not [71]. The presence of cross-cousin marriage is most often linked to the linguistic structure called *crossness*. Crossness is the presence of a linguistic distinction between opposite-gender siblings; for example: the child of a father's sister is a cross-cousin because a father's sister is a father's opposite-gendered sibling (or crossed-gendered sibling). When crossness occurs within the parental generation it is commonly called a bifurcate-merging terminology, which merges same-sex siblings, and splits (or bifurcates) the opposite sex siblings [59]. This often leads to two classes of relatives per gender: for male referents, father and father's brother are grouped under the same term, and mother's brother is a separate term (as mother's opposite-sex sibling), and mother and mother's sister are grouped, and father's sister is a separate term [72]. Historical interest in bifurcate-merging terminology surrounds two conditions. In the archetypal society with mandatory first-cousin cross-cousin marriage, a parent's opposite-sex sibling is also a spouse's parent (or a parent-in-law), making a kinterm applied to this relative polysemous, between a consanguineal and affinal relative (call this type A). Under these conditions, linguistic crossness could be seen to represent the limits of a culturally imposed incest taboo. However, bifurcate-merging terminology is often observed in societies lacking mandatory cross-cousin marriage, and thus a parent's opposite-sex sibling and parent-in-law are different people (type B). Determining why type B exists–i.e. why there should be crossness in the absence of mandatory cross-cousin marriage–is an ongoing topic of discussion amongst kinship terminology scholars [71, 73].

One suggestion is that all bifurcate-merging terminology occurs with some form of cross-cousin marriage, but does not require it to be mandatory for crossness to arise [71]. Here, we test the hypothesis that bifurcate-merging terminology co-evolves with allowable cross-cousin marriage in 56 Bantu societies (allowable being a weaker social norm than mandatory cross-cousin marriage). We define a language as bifurcate-merging if it distinguishes between a parent's same and opposite gender siblings. Specifically, we categorize a terminology as bifurcate-merging under three scenarios: if the term for father is merged with father's brother, and not mother's brother, if the term for mother is merged with mother's sister, but not father's sister, and if there is complete merging in both parent's male and female siblings (see S1 File for details). Cross-cousin marriage occurs in 38% of Bantu societies which have been studied, but the archetypal type A terminology is rare in contemporary languages, meaning a strict language-behavioral relationship is unlikely to be the main driver of crossness [50, 73]. Historical linguists have previously shown that the likely Proto-East-Bantu parental kinship terminology contained a bifurcate-merging pattern, from which a historical presence of preferential cross-cousin marriage is inferred [25], but it remains to be seen why bifurcate-merging terminology persists throughout the language family.

Cross-cousin marriage data are sourced from the Ethnographic Atlas via D-PLACE, and binary coded for the presence or absence of any cross-cousin marriage, and matched to Kinbank using Glottocodes [50, 74]. Language and marriage data are then paired to a sample of 100 Bantu phylogenies [75]. Using phylogenetic models of coevolution, implemented in Bayes-Traits v3.0.1, we fitted two different models, one where changes between the linguistic and behavioral traits depend on each other (co-evolve), and one where they are independent. We used Bayes factors to compare the likelihood of each model. A Bayes factor >2 indicates

positive support for the dependent model and is evidence for a coevolutionary relationship. All models were run twice to test MCMC convergence (see details in the S1 File).

**Example 2: Results.** We find that there is no positive evidence for the coevolution of cross-cousin marriage and bifurcate merging terminology in any of the three terminology variables. In tests of "complete bifurcate-merging", and "bifurcate-merging in women", there is no evidence either for or against the co-evolution of crosscousin marriage and bifurcate-merging organizations (Complete log Bayes Factor = -0.83; Women: -0.18). Tests of "bifurcate-merging in men" show positive evidence for independent evolution (-2.28), suggesting in this set of data, bifurcate-merging terminologies have no general relationship to cross-cousin marriage. Fig 3 displays the data for the "complete bifurcate-merging" variable on the Bantu phylogeny. The internal nodes, shown as probabilistic pie-charts for the four possible states, show confidence in relatively recent changes, but contain high levels of uncertainty in deeper nodes.

**Example 2: Discussion.** The relationship between bifurcate-merging terminology and cross-cousin marriage is statistically inconclusive from this analysis, adding to an increasing body of evidence disputing purported relationships between kinship terminology and social behaviors [29, 37]. In particular, it may be the case that societies can arrive at the same structural organization of kinship terminology but with categories conveying different cultural meanings. For example: Bena (Tanzania) speakers use linguistic crossness, allow but do not prescribe cross-cousin marriage, as well as practicing polygyny [76], while Lumasaaba (Kenya) speakers use crossness but no cross-cousin marriage [77]. Amongst Bena, the distinction of cross-cousins reflects the predicted relationships between marriageable and non-marriageable cousins. Cross-cousin marriage is considered a high-status marriage, although other marriages frequently occur. The emic category of cross-cousin in the Bena is broad and includes relatives stemming from a cross-cousin relationship many generations before (e.g. great great fathers' sister's offspring). The closer the genealogical relationship between the marrying couple, or if the relationship is traced through a person of high status, the more status is granted to the children of that marriage. In contrast, Lumasaaba parents' opposite-sex siblings play important roles in a child's life, by holding important ceremonial roles in a child's rite of passage and traditionally being the relative through whom the child will inherit wealth (although this tradition has since changed [77]). These important cultural roles highlight a special relationship between children and their mother's brother or father's sister, which could be the cause of the linguistic distinction. These are examples taken from our sample, but the relationship of kinship terminology structure to various semantic meanings is an avenue for future research.

Our coevolutionary results challenge a long-held belief within kinship terminology research. While some researchers have suggested that the relationship between kinship terminology and behavior is more complex than often assumed [78], the longstanding consensus of the field is that there is coupling between language and norms [1, 9, 31].

However, we test only one example of a behavioural-linguistic link from a sea of proposed hypotheses (e.g. [79, 80]), and we hope this database encourages future research on the relationship between kinship norms and language. Using Kinbank will aid in transparent and iterated hypothesis testing: as methods such as phylogenetic analysis become mainstream, and sophisticated statistical methods that incorporate not just shared cultural ancestry but spatial autocorrelation are developed, the granularity of kin term data in Kinbank allows the analyst to move flexibly between typological levels depending on the research question at hand.

## Conclusion

Kinbank offers an open and transparent database of kinship terminologies from a global sample of 1,229 languages. The examples presented in this paper illustrate the benefit of systematic

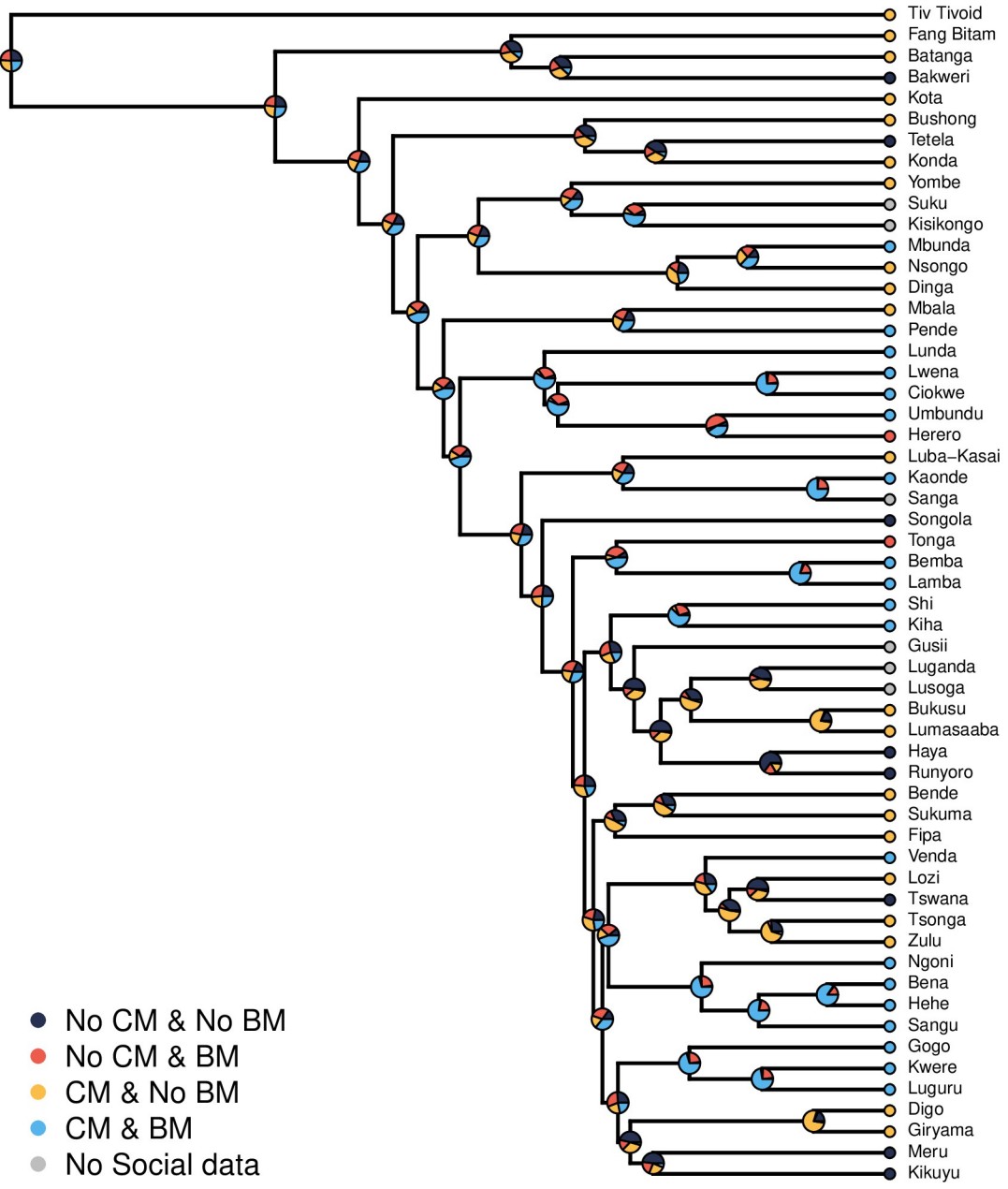

**Fig 3. Maximum clade credibility tree of Bantu languages displaying four possible states: No cross-cousin marriage (CM) and no bifurcate-merging (BM) of kinterms (black), no cross-cousin marriage with bifurcate-merging (yellow), cross-cousin marriage and no bifurcate merging (red), and both cross-cousin marriage and bifurcate merging are present (light blue), with their likely probabilities from a dependent model of evolution.** Languages marked with grey show where kinterms data were present, but social data were imputed by BayesTraits. Pie-graphs indicate the probability of states at each node, which were calculated using the BayesTrait command "RecNode". Deeper into the phylogeny there is much uncertainty as to the relationship between these two traits, indicated by the almost equal probability of all four states.

and flexible storage of large cultural datasets, as they combine the rich ethnographic record with decades of theoretical debate in linguistics and anthropology. Moreover, the focus on language family sampling allows for the application of computational phylogenetic approaches from evolutionary biology. While the cultural basis of kinship has long been an anthropological focus, interest from such fields as economics [7], psychology [27], and linguistics [31]

means that now, more than ever, there is a need for large, diverse cross-cultural datasets that allow us to test old and new hypotheses in one of the founding anthropological domains.

## Materials and methods

### Example 1: Are you my mama?

Kinbank contains 3,068 words for mother or father, from 1,022 languages. Some languages have multiple words for mother and father terms. All words are reduced to the first syllable of their root and coded for a consonant and vowel ignoring sound order, following [66]. The root syllable is typically the first syllable of the word, with exceptions being the presence of a prefix, or some other feature of the language which might indicate a different syllable. Prefix decisions are at the coders discretion. Syllables are coded as one of 34 consonant types and one of seven vowel types (following Blasi et al. 2016; S6 Table in S1 File). For example: the Serbian word for mother is *majka*: here the first and root syllable is [ma]. The first syllable [ma] has a consonant coded as [m] and vowel coded as [a]. Coding was performed manually and inter-rater reliability statistics are available in S7 Table in S1 File.

To control for the relatedness between languages, we use a global phylogenetic tree [81]. Though not a perfect measure of relatedness between languages, it is a vast improvement on correlational studies historically employed in kinship terminology studies (e.g. [9]). Pruning and cleaning of phylogenies was performed in *R* using the packages *ape* v5.3 [82], *phangorn* v2.5.5 [83] and *phytools* v0.7 [84].

**Model.** We built a phylogenetically-controlled repeated measures multilevel Bayesian logistic regression using brms v2.16.3 [85]. By using a repeated measures approach, multiple terms can be modelled per language (e.g. mother and father), while controlling for the phylogenetic relationships between languages. The response is a binary variable which indicates whether a term is mother (1) or father (0) and is independently predicted by the consonant and vowel sound codes. To control for language relatedness we included an inverse variance-covariance matrix built from the phylogenetic tree, as well as a random effect for language. This controls for both historical relatedness, and other factors that may be explained at the language level. Models were run for four chains, with 5,000 iterations, and 2,000 burn-in iterations. All fixed effects had normal priors with a mean of zero and standard deviation of 10. Probability estimates of consonant-vowel combinations of interest were predicted from the model posterior.

**Example 2: Does crossness indicate marriage preferences?.** We used terminological data from Kinbank to determine the presence of crossness across the whole parental generation (complete bifurcate-merging; F = FB $\neq$ MB & M = MZ $\neq$ FZ), or only within men (F = FB $\neq$ MB), or only within women (M = MZ $\neq$ FZ). Detailed data description is in S8 Table in S1 File. Cousin-marriage preference data are taken from the Ethnographic Atlas question EA023 from D-PLACE and coded to indicate the presence or absence of cross-cousin marriage (S9 Table in S1 File; [50, 74]), and linked to languages on the Bantu phylogeny (64). The intersection of Kinbank and D-PLACE results in a set of 56 societies for which 1) Kinbank contains kinterms for all parents and parents' siblings, 2) D-PLACE contains information on the presence of cross-cousin marriage and 3) the language is represented on a dated Bantu phylogeny. More information on data coding decisions is given in the S1 File.

We implement a Bayesian correlated evolution phylogenetic approach, using *BayesTraits* v3.0.1 [86]. By using a Bayesian approach with a sample of phylogenetic trees, the model does not only control for shared ancestry between societies but also for uncertainty in the phylogenetic relationships of languages. For each statistical test there are two models: one model where correlated evolution is assumed, and one where traits evolve independently. These

models are compared to calculate a log Bayes Factor (BF) to determine which model best fits the data. BF < 2 indicates weak evidence, > 2 positive evidence, 5–10 strong evidence, and >10 very strong evidence [87]. All models are run for 11,000,000 iterations, sampling every 1,000 iterations, with a burn-in of 1,000,000 iterations, on a posterior sample of 100 trees (approximately 200 samples per tree, but not enforced by using EqualTrees). All parameters have an exponential prior with a mean of 10. Each model was run twice to assess convergence; all MCMC runs show MCMC Gelman and Rubin statistics are <1.1 (S10 Table in S1 File). MCMC trace plots are available in S1 Fig in S1 File.

## Supporting information

**S1 File. Supporting information.** This file contains additional information on data collection and the methods used in the two examples.
(DOCX)

## Acknowledgments

We would like to thank Isobel Clifford, Lieke Hoenselaar, and Maarten van den Heuvel for their assistance with data collection. We would also like to thank two anonymous reviewers for their comments on the manuscript.

## Author Contributions

**Conceptualization:** Nicholas Evans, Fiona M. Jordan.

**Data curation:** Sam Passmore, Wolfgang Barth, Simon J. Greenhill, Kyla Quinn, Catherine Sheard, Paraskevi Argyriou, Joshua Birchall, Claire Bowern, Jasmine Calladine, Angarika Deb, Anouk Diederen, Niklas P. Metsäranta, Luis Henrique Araujo, Rhiannon Schembri, Jo Hickey-Hall, Terhi Honkola, Alice Mitchell, Lucy Poole, Péter M. Rácz, Sean G. Roberts, Robert M. Ross, Ewan Thomas-Colquhoun, Nicholas Evans, Fiona M. Jordan.

**Formal analysis:** Sam Passmore, Wolfgang Barth, Simon J. Greenhill.

**Funding acquisition:** Nicholas Evans, Fiona M. Jordan.

**Investigation:** Joshua Birchall.

**Project administration:** Sam Passmore, Wolfgang Barth, Simon J. Greenhill, Kyla Quinn, Catherine Sheard, Joshua Birchall, Terhi Honkola.

**Software:** Simon J. Greenhill.

**Supervision:** Nicholas Evans, Fiona M. Jordan.

**Validation:** Simon J. Greenhill.

**Visualization:** Sam Passmore.

**Writing – original draft:** Sam Passmore, Catherine Sheard, Nicholas Evans, Fiona M. Jordan.

**Writing – review & editing:** Sam Passmore, Wolfgang Barth, Simon J. Greenhill, Catherine Sheard, Paraskevi Argyriou, Joshua Birchall, Claire Bowern, Terhi Honkola, Robert M. Ross, Nicholas Evans, Fiona M. Jordan.

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
