## [Decision Letter · Decision Letter 0]

8 Sep 2022

PONE-D-22-20965Kinbank: A global database of kinship terminologyPLOS ONE

Dear Dr. Passmore,

Thank you for submitting your manuscript to PLOS ONE. After careful consideration, we feel that it has merit but does not fully meet PLOS ONE’s publication criteria as it currently stands. Therefore, we invite you to submit a revised version of the manuscript that addresses the points raised during the review process.

I sincerely appreciate the opportunity to review the manuscript. I have now received reviews from three experts in the area, who all have experience constructing large comparative databases, and I now feel that I can make an editorial decision at this time.

All the reviews were positive and constructive, and are advising revision.  The manuscript is well written, and provides a good outline of a resource that will likely be important to a multitude of disciplines.  Remarks from all reviewers are clear, so I will not repeat everything here. Please provide responses to all reviewer comments. The issues of particular importance for your revisions are as follows:

Reviewer 1 raises concerns about the representativeness of the sample presented in the manuscript. Please address these concerns and questions, and provide further discussion of sample selection and any potential associated biases.Reviewer 2 makes an important request for further technical details about how you constructed the database. This information will certainly be essential for researchers to use the resource with ease, and given the amount of effort that was no doubt needed during database design and construct, you should showcase this hard work!  The reviewer's comments are detailed and clear so I will not labour this, but please do provide detailed responses to all of the reviewers points and questions. 

We look forward to receiving your revised manuscript.

Kind regards,

Daniel Redhead

Academic Editor

PLOS ONE

Journal Requirements:

"FMJ received funding from the European Research Council (Starting Grant VARIKIN ERC-Stg-639291). NE received funding from the Australian Research Council Center of Excellence for the Dynamics of Language (Grant CE140100041). JB and FMJ received funding from the Leverhulme Trust (International Partnership Mobility Grant 160281)."

4. Please amend the manuscript submission data (via Edit Submission) to include author "Lucy Poole".

5. We note that Figure 1 in your submission contain map images which may be copyrighted. All PLOS content is published under the Creative Commons Attribution License (CC BY 4.0), which means that the manuscript, images, and Supporting Information files will be freely available online, and any third party is permitted to access, download, copy, distribute, and use these materials in any way, even commercially, with proper attribution. For these reasons, we cannot publish previously copyrighted maps or satellite images created using proprietary data, such as Google software (Google Maps, Street View, and Earth). For more information, see our copyright guidelines: http://journals.plos.org/plosone/s/licenses-and-copyright.

Reviewers' comments:

Reviewer's Responses to Questions

**Comments to the Author**

1. Is the manuscript technically sound, and do the data support the conclusions?

Reviewer #1: Yes

Reviewer #2: Partly

Reviewer #3: Yes

2. Has the statistical analysis been performed appropriately and rigorously? 

Reviewer #1: Yes

Reviewer #2: Yes

Reviewer #3: Yes

3. Have the authors made all data underlying the findings in their manuscript fully available?

Reviewer #1: Yes

Reviewer #2: Yes

Reviewer #3: Yes

4. Is the manuscript presented in an intelligible fashion and written in standard English?

Reviewer #1: Yes

Reviewer #2: Yes

Reviewer #3: Yes

5. Review Comments to the Author

Reviewer #1: This will be a very useful dataset for many social scientists. I wish I had access to such a dataset years ago. I believe the paper meets the standards of the journal.

The dataset is constructed carefully and kinterms make sense when I checked them for the few languages which I am familiar with.

However, I have a few small comments and suggestions:

• My main concern is that the sample of 1,168 languages (out of 7,560 live languages according to the Ethnologue) might not be a representative sample of the world population. For example, if you are relying on the politically dominant languages of countries (which are more likely to have dictionaries), there will be a selection bias in your sample.

The abstract says “representative sample”, but there is no discussion of that in the paper. What percentage of world population and each continent is covered by the sample? You can use Ethnologue maps and population data to give us some estimates and convince us that this is indeed a representative sample.

It would be helpful if you tell us why thousands of live languages are no included in the sample. Not all readers are familiar with your data collection restrictions.

• It would be interesting to see example 2 presented with cousin marriage preference variable (EA025) as well. Some argue that cousin marriage preference better reflects the actual marriage practices of societies than cousin marriage permitted.

According to Jack Goody (1976)’s analyses, there is a strong correlation between inheritance, cousin marriage, and Eskimo and descriptive categories of cousin terms. I am not aware of later literature on Goody’s hypothesis or analyses, but it might worth considering it as an additional example.

• It might be useful to mention that your dataset can be linked to today’s many surveys. Giuliano & Nunn (2018) and Bahrami-Rad, Becker, Henrich (2021) provide methods to link EA to survey data. The later provides the data on a website: http://dgce.fas.harvard.edu/. Despite its all shortcoming, the method seems to be becoming popular among economic historians.

References:

Goody, Jack, and John Rankine Goody. Production and reproduction: a comparative study of the domestic domain. No. 17. Cambridge University Press, 1976.

Giuliano, Paola, and Nathan Nunn. "Ancestral characteristics of modern populations." Economic History of Developing Regions 33.1 (2018): 1-17.

Bahrami-Rad, Duman, Anke Becker, and Joseph Henrich. "Tabulated nonsense? Testing the validity of the Ethnographic Atlas." Economics Letters 204 (2021): 109880.

Reviewer #2: In this article, the authors introduce a new database of the diversity of terms with which speakers of a large number of languages refer to their genealogical kin. After a short introduction, they present two example analyses for the potential that this database might offer for future research into the links between language, cognition, and behavior. My main problem with the article in the current version is that I don't think there is sufficient information on how the database was built, which supposedly is the key point of this article. Given that the aim is to encourage people who have thus far not been involved in this collaborative effort to built the database to use it, I think this information is crucial for readers to judge whether to access the data. A problem of working with a database that someone else has put together is that biases might come in if a full understanding of the database is lacking. For example, it seems like you currently would not advise an analysis into why certain terms are missing from specific languages? Or that some global analyses might not show any patterns because of "Simpson's paradox" of diverse patterns within different regions? The complication is that usually data are collected to answer a specific question, or with a specific framework in mind. It appears that also was the case here before you started to combine efforts and build this broad database. This introduces restrictions on what the data can and cannot be used for that are not always immediate obvious to outsiders. I this adding more details on the decisions you made when building the database can help the readers understand whether the information is relevant for the questions they are interested in. I think most of the information should be part of the main manuscript, but any further information you think could be of relevance would also be welcome in the supplementary materials (I also checked the associated websites and did not find this information). None of my comments invalidate any of the effort, and, independent of the replies and changes, I think this article has the potential to be of interest to many readers.

First, there are some broad questions:

How was the database actually built? Is the information all from secondary literature, or did it involve primary data collection for some of the languages of anthropologists asking native speakers about the terms and their meaning? If it is based on secondary literature, how do you account for the potential risk that original encoders might have had a particular focus (either because of their research focus or their own cultural background) such that they might not have recorded all relevant kin terms? You mention in the supplementary materials that one potential source of disagreement between the different databases is that they relied on different sources. Did you have a criterion to say that you thought a given language is accurately represented or did you simply enter all information you might have found in a representative source? In such cases, did you note that the information might be incomplete?

I recognise that you provide details on each entry, but I think it is necessary to provide information on the general rules here in the manuscript.

You focus on the challenge that occurs when a kin term is absent and you leave a cell empty, but what information is lost in the opposite case where an additional term might exist in a language that encodes a relationship beyond that stored in the 115 types you focus on?

What was the process through which you came up with the 115 kin types? This seems like it was a critical step in the design of Kinbank, so I think additional information should be provided here. Was this an iterative process? Did you have the genealogical relationships decided a priori? Providing this information will help others to think about what these types might mean, and how they might want to classify relationships

Given that the focus is on the database, maybe some brief descriptive summary of the content could be helpful before you jump into the example analyses. For example, is there any language in which there are separate terms for all these 115 kin types? What is the average number of kin terms per language?

What does a kappa value of 0.80 signify for inter-rater reliability? Does kappa range between 0-1, with higher values indicating higher agreement? Does this mean that roughly 20% of terms were classified differently between the two raters? Do you think these are systematic biases, where coders primarily familiar with a certain system/group of languages might interpret information differently than coders trained on a different system/language? For the reliability, did you also have native speakers (or linguists) spot-check entries to ensure that the information you transcribed accurately reflects the kin terminology of their language?

Some more specific question:

Are the emic, language specific, connotations recorded and provided in Kinbank?

Why are Sino-tibetan languages not included?

Is there a way for people to contribute?

Additional comments:

The second paragraph of the introduction with some examples of kin terminology appears written from an English-speaker perspective, with the assumption that the corresponding kinship terminology will be automatically in the readers mind. As a non-native English speaker, I could follow along, but it might be helpful to phrase the examples in broader context (as you yourself point out in the methods) by always contrasting two or more systems.

To reach the people who apparently have been using outdated concepts based on limited data, it might help to broaden the final paragraph of the introduction by referring what the potential use of the database in different fields of research might be. Currently you focus on specific examples of research that has already been done, which is maybe more limited in scope than what you envision the full potential of the database to be. The examples are also not always immediately accessible to outsiders, for example, it was not clear to me what "a collection of fresh field research alongside new methods of Tupian and Cariban kinship" means and why and for whom this matters. Linked to this, the final sentence of the introduction focuses on "questions on the origin and maintenance of cross-cultural variability in kin categorization", whereas before you seem to have a broader scope in also highlighting studies that investigate how language might guide behavioural norms.

I think it would be helpful if, in the introduction, you have a sentence each about the two example analyses you performed.

Again, given the broad audience this article might reach, I think it could help to provide some brief explanations for some of the analytical choices you made in the two examples. For example, in example 1, why did you use a phylogenetic approach? Naively, the question could be interpreted as a "human bias", so maybe we are simply interested in how many people use these terms? Maybe briefly explain what issue the "regional and historical relationship" introduce such that it is important to "control for linguistic history in comparative hypothesis tests".

In example 1, how many consonants and how many vowels were there? Only the five that are in figure 2? The supplementary material suggests that there are many more. Is the low informativeness of vowels because there are so few of them?

Reviewer #3: The paper is well written, presents an important dataset for studies of kinship and cultural evolution, and uses it to assess two important sets of hypotheses. In the future it would be useful to explore reasons for disagreement in the initial reliability testing and perhaps refine the dataset using multiple coders across all languages in the dataset to reduce random and systematic sources of error in the key variables.

6. PLOS authors have the option to publish the peer review history of their article (what does this mean?). If published, this will include your full peer review and any attached files.

Reviewer #1: No

Reviewer #2: No

Reviewer #3: No

---

## [Decision Letter · Decision Letter 1]

6 Mar 2023

Kinbank: A global database of kinship terminology

PONE-D-22-20965R1

Dear Dr. Jordan,

We’re pleased to inform you that your manuscript has been judged scientifically suitable for publication and will be formally accepted for publication once it meets all outstanding technical requirements.

Kind regards,

Daniel Redhead

Academic Editor

PLOS ONE

Additional Editor Comments (optional):

This is a great paper, and reviewers and myself believe that it is now ready for publication. The authors have addressed all comments made by the reviewers and myself. Please do take into consideration the final minor comments made by reviewer 2 when resubmitting the manuscript. 

Reviewers' comments:

Reviewer's Responses to Questions

**Comments to the Author**

1. If the authors have adequately addressed your comments raised in a previous round of review and you feel that this manuscript is now acceptable for publication, you may indicate that here to bypass the “Comments to the Author” section, enter your conflict of interest statement in the “Confidential to Editor” section, and submit your "Accept" recommendation.

Reviewer #1: All comments have been addressed

Reviewer #2: All comments have been addressed

2. Is the manuscript technically sound, and do the data support the conclusions?

Reviewer #1: Yes

Reviewer #2: Yes

3. Has the statistical analysis been performed appropriately and rigorously? 

Reviewer #1: Yes

Reviewer #2: Yes

4. Have the authors made all data underlying the findings in their manuscript fully available?

Reviewer #1: Yes

Reviewer #2: Yes

5. Is the manuscript presented in an intelligible fashion and written in standard English?

Reviewer #1: Yes

Reviewer #2: Yes

6. Review Comments to the Author

Reviewer #1: (No Response)

Reviewer #2: Thank you for addressing my previous comments so diligently and for adding the further details to your manuscript. I think this article is now a great introduction to this highly valuable resource.

For the final manuscript, you might want to consider the following minor points:

- In line 162 you state that "The Kinbank database is freely accessible" - is there a licence attached for access and re-use (e.g. OpenData- or Creative- Commons Licence)? It might be worth mentioning the licence here to clarify how others can access the data.

- From your description it also appears that your database falls within the FAIR data principles in case you want to mention this.

- There is a break in the section on The Kinbank sample between the first and second paragraphs (line 220) because it is not immediately clear whether Parabank, Varikin etc. are alternative database to which you compare Kinbank or whether these databases have been integrated into Kinbank. You mention this in an earlier section, but it might help to repeat here that Kinbank results from the merging of these aligned research projects.

- In your reply, you clarify that there are 155 core kin types, but in some places of the manuscript you still seem to refer to the 115 kin types that were originally listed. For example, in lines 262-263 the 88 geneaological kin and 27 kin by marriage add up to 115, not 155; and the references Table S2 appears to only contain 98 entries.

7. PLOS authors have the option to publish the peer review history of their article (what does this mean?). If published, this will include your full peer review and any attached files.

Reviewer #1: No

Reviewer #2: No

---

## [Editor Report · Acceptance letter]

26 Apr 2023

PONE-D-22-20965R1 

Kinbank: A global database of kinship terminology 

Dear Dr. Jordan:

I'm pleased to inform you that your manuscript has been deemed suitable for publication in PLOS ONE. Congratulations! Your manuscript is now with our production department. 

Kind regards, 

on behalf of

Dr. Daniel Redhead 

Academic Editor

PLOS ONE